# First Steps of Asthma Management with a Personalized Ontology Model

Hicham Ajami [1], Hamid Mcheick [1,*] and Catherine Laprise [2]

1 Department of Computer Science and Mathematics, Université du Quebec à Chicoutimi, Chicoutimi, QC G7H 2B1, Canada; hicham.ajami1@uqac.ca
2 Fundamental Sciences Department, Université du Quebec à Chicoutimi, Chicoutimi, QC G7H 2B1, Canada; catherine_laprise@uqac.ca
* Correspondence: hamid_mcheick@uqac.ca; Tel.: +1-418-545-5011 (ext. 5676)

**Abstract:** Asthma is a chronic respiratory disease characterized by severe inflammation of the bronchial mucosa. Allergic asthma is the most common form of this health issue. Asthma is classified into allergic and non-allergic asthma, and it can be triggered by several factors such as indoor and outdoor allergens, air pollution, weather conditions, tobacco smoke, and food allergens, as well as other factors. Asthma symptoms differ in their frequency and severity since each patient reacts differently to these triggers. Formal knowledge is selected as one of the most promising solutions to deal with these challenges. This paper presents a new personalized approach to manage asthma. An ontology-driven model supported by Semantic Web Rule Language (SWRL) medical rules is proposed to provide personalized care for an asthma patient by identifying the risk factors and the development of possible exacerbations.

**Keywords:** asthma; SWRL; medical rules; pervasive computing; ontology model; architectural model; chronic respiratory disease; management systems

## 1. Introduction

Asthma is a chronic respiratory disease in which the lining of the lung airways becomes swollen and narrowed thus reducing the airflow into and out of the lung. This airflow obstruction can cause symptoms such as coughing, wheezing, shortness of breath, and chest tightness. The severity and the frequency of asthma exacerbations differ from one patient to another [1,2]. Asthma usually appears during childhood; however, some adults can also develop asthma [3]. Estimations show that asthma affects more than 339 million people worldwide [2]. In Canada, more than 3.8 million individuals [4], who are over one year of age, are asthmatics. Asthma can affect the quality of life of an asthma patient as well as his/her work productivity and psychological health [1]. There are two phenotypes of asthma: extrinsic or allergic asthma and intrinsic or non-allergic asthma. Allergic asthma is triggered after the inhalation of specific allergens: it usually develops early in life and is usually accompanied or preceded by atopic diseases such as allergic rhinitis, a food allergy, and atopic eczema [5,6]. Atopy is the genetic susceptibility of an allergic reaction being provoked after an exposure to an allergen [7]. On the other hand, in non-allergic asthma, no exogenous allergens can be identified, and this phenotype is characterized by a late onset. Allergic asthma is considered to be more responsive to treatment than non-allergic asthma, and it is the most common form of the disease [8]. An asthmatic response is provoked by an allergy in 75–80% of all asthmatic cases [9–11]. Asthma exacerbations are often triggered upon the exposure to environmental allergens or irritants (e.g., indoor and outdoor allergens and pollution), tobacco smoke, workplace irritants (e.g., food derivatives, fumes, chemical products, animal products, etc.), changes in the weather conditions or viral respiratory infections [1,2,6,7,9].

Over the past decades, asthma problems have been of particular interest in bioinformatics projects. However, asthma is a complex disease where multiple aspects need to be tackled to control its complications. Each asthma patient experiences such conditions uniquely; therefore, dynamic protection standards must be developed for each patient. Asthma patients need constant care to protect them from various environmental irritants and allergens. For example, dust mites, animal dander, humidity, extreme temperature, precipitation, some weed pollens and molds, and air pollutants [2,7,9], such as particulate matter $PM_{2.5}$, $PM_{10}$, $O_3$, $NO_2$, $SO_2$, and CO, pose potentially serious and life-threatening risks to asthma patients. On the other hand, food allergies can increase the bronchial hyperreactivity in asthmatic patients as a part of the anaphylactic reaction. Although a food allergy is triggered after the ingestion of certain food derivatives, inhaled airborne allergens such as flour, egg allergens, nuts, soybean, tea dust, fish, and seafood may induce asthmatic reactions [6].

This paper proposes a first step to design an ontology-based approach to evaluate the health status of asthma patients. The originality of the proposed approach resides in the dynamic control of thresholds that govern the triggers of asthma. This paper builds a formal knowledge base of the relevant parameters and their relationships that can be used to monitor the environmental risk factors and food allergens depending on the medical profile of the patient. The remainder of the paper is structured as follows. In Section 2, we review existing tele-monitoring systems and ontology-based models in the asthma domain. Section 3 describes our expert rules ontology to deal with asthma. Section 4 implements the system and evaluate its result. Finally, this work is concluded in Section 5.

## 2. State of the Art Systems

The domain of digital health has grown rapidly over the last ten years. Healthcare projects are moving towards tele-monitoring systems, patients' self-management, and computer-mediated counselling. The reports about chronic diseases have shown the need for a variety of effective solutions to improve the lives of patients. Context awareness and personalization are the most important elements of the success of cognitive computing in the modern healthcare domain. Context awareness refers to the use of external information that can influence a person's situation, while personalization provides a tailored treatment approach to address each patient's condition. Personalized context awareness can be realized using an ontological model for abstraction. The ontology that could be defined as a formal description of knowledge is one of the common methods to deal with the challenges of designing, managing, and integrating patients' health data from heterogeneous sources to extract and infer useful information. Cognitive computing is currently being used in several asthma projects, for instance, Quinde et al. [11] presented an approach to develop context-aware systems aiding the individualized management of asthma. This work strives to shed light on the existing gaps of using context awareness in asthma management and determine the functionalities of such a context-aware system. Al-dowaihi et al. [12] propose an asthma prototype system that allows patients to self-supervise and manage their symptoms and conditions accurately in air-polluted areas, as well as informing their healthcare providers in critical cases.

Kwan et al. [13] developed a portable external mobile device accessory to collect PEF, FEV1, FEV6, NO, CO, and $O_2$ from patients. The authors developed an application to record this information and send the results to a physician to track asthma symptoms and lung function in real time. According to the authors, this work would allow the physician to make an appropriate intervention in a patient's medication regimen more quickly. Anantharam et al. [14] created a system called kHealth to aggregate multisensory and multimodal data from asthmatic patients using a combination of active and passive sensors. The project presents an advanced data analysis platform that can help physicians determine more precisely the cause and severity of asthma and therefore improve the quality of life of patients. Ra et al. [15] proposed a cloud-based system called AsthmaGuide, in which a smartphone is used as a center point for gathering information in real-time processes.

The data are then uploaded to a cloud web application for both patients and doctors to receive advice and alarms. AsthmaGuide allows asthma patients to be involved in their care and treatment and allows healthcare professionals to provide more effective support. Dieffenderfer et al. [16] developed a wearable sensor system that enables us to measure the correlation between individual environmental exposures and physiological markers and subsequent adverse reactions. This framework will grant us understanding of the impact of increased ozone levels and other pollutants on asthma. Gyrard et al. [17] developed a personalized healthcare system for chronic diseases such as obesity and asthma; this system aggregates knowledge from different sources such as internet of things devices, clinical documentation, and electronic health records. Quinde et al. [18] proposed a new context-based approach to control asthma; this solution follows personalized management to address the heterogeneity of asthma. In addition, Galante et al. presented an approach to self-monitoring guidance (see Asthma management resources for healthcare) but a dynamic aspect is not implemented [19]. Singhal et al. propose also a Context Awareness for Healthcare Service Delivery in asthma domain but they focus mainly on sensor capacity [20].

Many other researches have been conducted in the asthma field. Early life sensitization to indoor allergens is a predictor of asthma development later in life. Furthermore, the avoidance of exposure to these allergens continues to be important, especially given that the vast majority of children with asthma are sensitized to at least one indoor allergen [21]. Indoor allergens are of particular importance and principally include house dust mites, animal dander, cockroaches, mice, and molds [21,22]. The relative importance of these different allergens varies based on different environmental factors, depending on geographic, climatic, socioeconomic, and housing conditions. In the outdoors, pollens can induce seasonal asthma in sensitized individuals, and outdoor molds or fungi may lead to severe asthma exacerbations [23]. Changes in gaseous and particulate outdoor air pollutants are associated with daily asthmatic symptoms, a decrease in lung function, emergency room visits, and hospitalizations for asthma attacks [23]. Several studies have confirmed that air pollution from ozone ($O_3$), sulfur dioxide ($SO_2$), nitrogen dioxide ($NO_2$), and particulate matter (PM) may induce or aggravate asthma [24]. The most important outdoor air pollutants are PM, $O_3$, $SO_2$, $NO_2$, CO, and Lead (Pb). Exposure to environmental tobacco smoke (ETS) in early life, especially that from the mother, and maternal smoking during pregnancy, are known risk factors for respiratory symptoms and asthma among children. Active smoking has been shown to be risk factor for developing asthma; women who smoke are at particular risk. Similarly, second hand tobacco smoke (SHS) exposure is also associated with the development of asthma in adolescents and adults [8,25]. The workplace environment can lead to the development of different types of work-related asthma. Occupational asthma (OA) results from the exposure to irritants and allergens in the workplace, such as food derivatives, fumes, gases, chemical products, animal products, etc. [8,26]. Cold weather causes functional disabilities among individuals with an existing respiratory disease. This is because low temperatures and the accompanying low air humidity are likely to affect the respiratory epithelium and induce hyperresponsiveness and narrowing of the respiratory airways [27]. Cold temperatures can trigger asthma attacks. In general, the effect of cold weather appears to last for several weeks, whereas the effect of hot weather is more short term [23].

These research projects handle, in particular, the heterogeneity, yet, despite their medical value, they lack many aspects that may affect the lives of asthma patients. The developed frameworks and applications do not adequately support a comprehensive control of asthma either in terms of the indicators to be tracked or in terms of physical location changes. None of these research projects provide a detailed protection platform to monitor all relevant environmental triggers and food allergens. For example, a combination of the context categorization, formal context representation, and reasoning is not yet used. In addition, these frameworks need a personalized treatment asthma plan since each patient reacts differently to these triggers. In this research work, we aim to fill the gap by proposing

a personalized healthcare system to better protect asthma patients from the environmental risk factors and food allergens.

## 3. System Architecture

The proposed system is a context-aware architecture that is able to infer events and provide real time recommendations. The system aims to manage the environmental and nutritional impacts and reduce the implications of allergic triggers according to the medical profiles of patients. As shown in Figure 1, this system is composed of four layers: a data acquisition layer, a representation layer, a reasoning layer and an application layer.

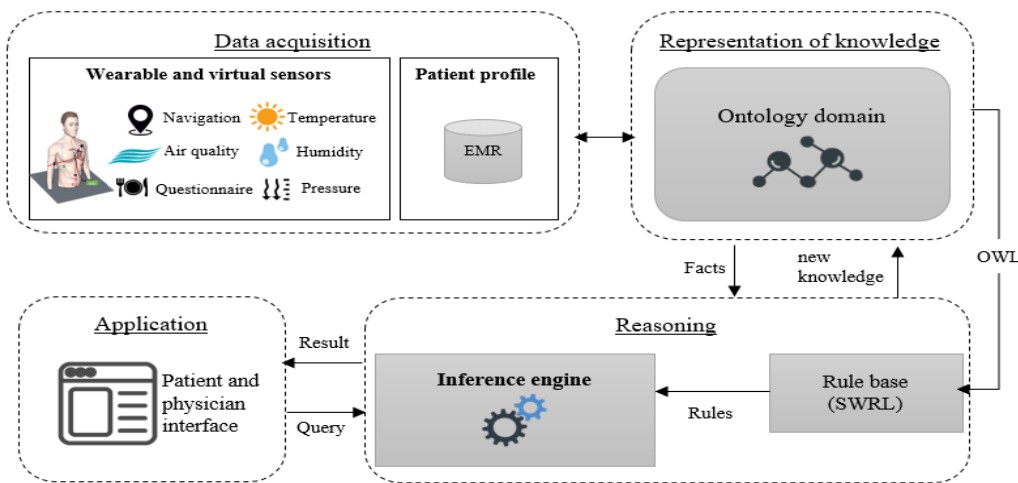

**Figure 1.** Proposed system architecture.

The data acquisition layer is aimed at collecting the medical profile of asthma patients, foods, and environmental information from active or passive sensors. The representation layer is an ontological schema that has been used to translate the gathered information into machine understandable data. The reasoning layer is the brain of this system where all the inference processes are carried out through predefined medical rules. The application layer includes the functions and services offered by the system. All these layers are discussed in further detail in the following sections.

### 3.1. Data Acquisition

The acquisition layer is primarily used for collecting and preparing asthma surveillance data. As we mentioned above, this part consists of the patient's medical profile as well as the nutritional regime (i.e., food) and environmental exposure (i.e., air quality and weather conditions) in real time. We rely on four core components in the acquisition process, the medical profile information is obtained from electronic health records, while the individual's location, daily intake nutrients, and environmental real-time data are collected through sensors and the web services interface. The historical medical profile contains the demographic characteristics (i.e., age, sex, BMI, occupation, city, etc.), lung function tests, asthma level, biomarkers, comorbidities (personal and family health history of chronic diseases), treatment, smoking habits and exposure to second hand smoking, allergy testing for animals, herbs, dust, and molds, and food. Environmental factors (both ingested and inhaled) have been suggested to contribute to asthma pathogenesis. Examples of environmental factors include indoor allergens, outdoor allergens, air pollutants, respiratory viruses, extreme weather, tobacco smoke, and chemical irritants in the workplace (occupational risk factors) [2,7,9]. This information (e.g., temperature, humidity, ozone, carbon dioxide, carbon monoxide and nitrogen dioxide, particulate matter, and sulphur dioxide, etc.) can be detected through air quality sensors that monitor the presence of asthma triggers in the surrounding area. On the other hand, location sensors (GPS) can

be used to infer an individual's daily mobility. Observational studies and experiments have shown that components of foods can influence the status of patients' lives, where many essential nutrients can increase the risk of developing asthma symptoms. Hence, the nutrition program for patients must be monitored continuously through a questionnaire that is filled out electronically on the user-friendly platform.

### 3.2. Representation Layer

The representation layer provides a structured knowledge description for real-time situations from the acquired massive data. Essentially, the data representation layer includes an ontology model that describes the disease, environment, equipment, patient profile, and allergens. This ontology is a first step towards a complete representation knowledge of asthma, and it will be constructed with whole sub-ontologies in the future research work. This ontology extends and reuses the ontology that we proposed for chronic obstructive pulmonary disease (COPD). The developed ontologies consist of concepts related to the personal and medical profile, physical examinations, laboratory tests, location, activity, environment, time, recommendations, and diseases. In this work, we developed an ontology with a focus on the relation between diet and environment with the patient's health. We based the construction of our ontology on the steps of Sanchez [28]: (i) determine the domain scope, (ii) ontology reuse, (iii) the development of a conceptual ontology, (iv) implementation, and (v) evaluation. In addition, the development of our ontology was driven by the following main questions:

- Which information are necessary for having a detailed description of each context?
- Which concepts are needed for supporting the design of medical rules allowing patient monitoring?
- Which data have to be provided by patients to enable reasoning processes?

Our ontology construction based on Sanchez's approach is summarized in Table 1.

**Table 1.** Sanchez's approach to design ontology.

| Step | Approach |
| --- | --- |
| 1 | Identification of the domain and scope of the ontology, asthma domain, and alert management. |
| 2 | Ontology reuse and addressing poor ontological coverage of pulmonary diseases such as asthma. |
| 3 | Development of a conceptual model. |

At this level, the proposed ontology is mainly related to fundamental ontologies presented in COPDology, BioPortal, FoodOn, and SNOMED-CT [29]. These ontologies can be easily combined to cover further domains such as asthma disease. Thus, our asthma ontology is an integration of several ontologies relevant to interpreting the risk factors:

- Asthma from BioPortal
- Weather and environment ontologies from COPDology [30]
- Food allergens from FoodOn
- Symptoms and pollen concepts from SNOMED-CT

On the other hand, the relevant information and entities (parameters) can be identified using our process [31]. These parameters should be offered by the ontology to monitor the allergic issues.

Figure 2 shows a general overview of the main concepts that can be found in our ontology. In the next paragraphs, we distinguish the concepts into four different categories and we provide the semantic significance of the principal entities.

In connection with food allergens, FoodOn classes are referenced via the "has allergic" relation. FoodOn classes describe food entities and their associated molecules hierarchically. "Food" is the root concept of all foods contained in this ontology. As subclasses of this concept, we defined three main subconcepts: "Molecular Compounds", "Group", and "Quantity"; with these three subclasses, we can provide a full description of food properties and daily serving amounts. In addition to these concepts, many characteristics have been

defined. Indeed, modelling such knowledge has been created to facilitate the task of patient monitoring. For example, based on the patient profile, healthcare providers may decide to define fine-grained monitoring on specific nutrients. For instance, patients with gluten intolerance issues should limit the consumption of wheat, barley and rye, or people having nut allergies should avoid nut butter, almonds, cashews, peanuts, and others. The location concept concerns detailed information about the geographical area that can be exploited for different monitoring purposes. For example, when pneumologists create a rule, they might suggest directive intended for a specific place or a given period. By considering the location, more protective procedures are presented by the sub-concepts "Indoor", "Outdoor", "Home" and "workplace". The asthma patient is represented in our ontology for linking purposes.

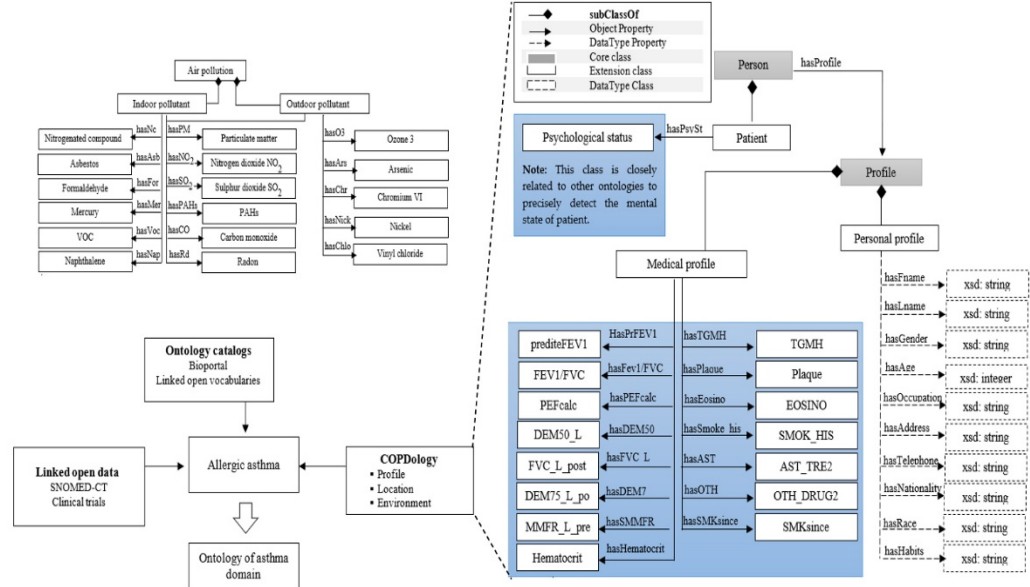

**Figure 2.** Part of our ontology model.

The patient's medical concepts are imported from SNOMED-CT, e.g., the stage of the disease, symptoms, lab tests and diagnostic procedures. Within this ontology, we modelled the "Patient" concept as a bridge between the food ontology, location ontology, environment ontology, and asthma ontology. The environmental concepts that trigger asthma attacks include six main subconcepts: "Indoor Allergen", "Outdoor Allergen", "Ambient Air, "Weather", "Smoking" and "Pollution". The indoor allergen class principally includes house dust mites, animal dander (e.g., cat, dog, horse), mice and molds. In the outdoor allergen class, we modelled pollen, molds and fungi concepts that can induce seasonal asthma and may lead to severe asthma exacerbations. The measurement of indoor and outdoor allergens provides a consistent basis for determining the risk associated with the development of an asthma attack. In addition to these categories, the environment contains ambient air, weather, smoking and pollution. Each of these entities branches into dozens of sub-elements. For example, ambient air is a root class of the hierarchy composed of $N_2$ and $O_2$, $CO_2$, argon, neon, hydrogen, methane, xenon, krypton and helium. The pollution concept, on the other hand, includes a wide variety of air pollutants such as PM, $O_3$, $SO_2$, $NO_2$, CO, and lead (Pb). Smoking is divided into "Active smoking" and "Second hand smoke". Active smoking is a risk factor for developing asthma. Similarly, exposure to second hand smoke is also associated with the development of asthma complications. Asthma ontology concepts are imported from BioPortal. Asthma represents the main root of all classes and it comprises a stage, treatment, risk factors, conditions, and characteristics, etc.

Evaluation of ontology: There are many criteria and metrics to evaluate ontology. These characteristics measure the complexity of our ontology on both class and ontology

levels. Ontology engineers compare their structure against a "gold standard" reference. The evaluation values presented in Figure 3 belong to some medical ontologies [31–39]. At the level of ontology, the size of vocabularies (SOV) of our ontology exceeds 1800 concepts. As a first step of our ontology, the edge node ratio (ENR) of our work is more useful than the other ontologies, but, in fact, our ontology could be more complex. The TIP value is a rational indicator of inheritance relationships. The TIP value of our ontology reaches 20, which means that this inheritance hierarchy deviates significantly from the tree root. The entropy of the ontology graph (EOG) corresponds to the distribution of relations. The EOG of our ontology is almost 2, thus it has a relatively acceptable structure. At the class level, the number of classes (NOC) and the number of instances (NOI) are 180 and 4000, respectively. The high number of properties (NOP) supports the reasoning engine [40]. The number of root classes (NORC) in this ontology refers to the great diversity found in this structure. The high value of AP is a good indication that our semantic representation has enough information. CR is 90, which means that most of the classes have instances. This description of knowledge is very rich, where its RR is equal to 40. The high inheritance richness (IR) reflects the vertical and detailed nature of representation. For more details, please refer to [30].

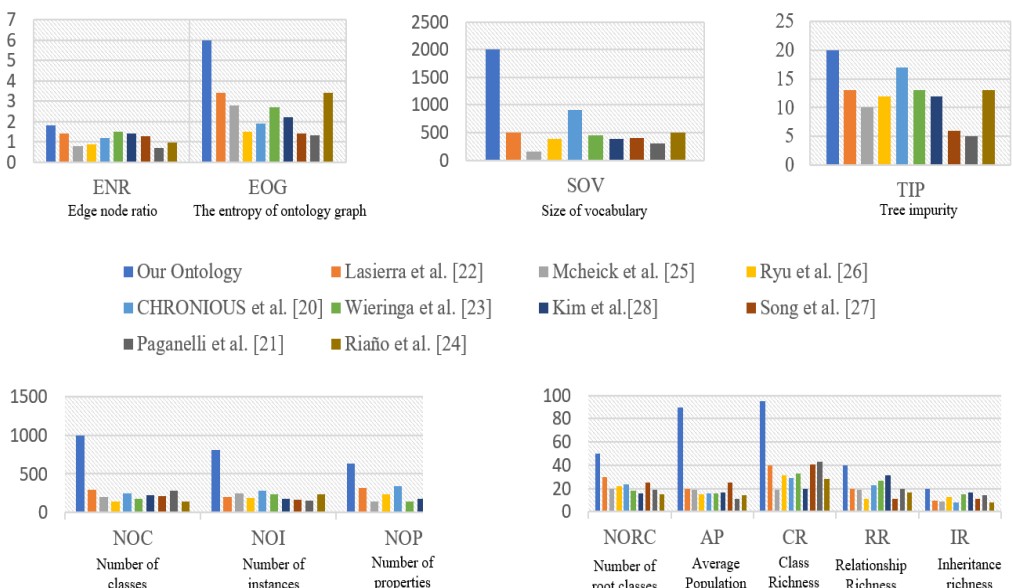

**Figure 3.** Design assessments.

### 3.3. Reasoning Layer

The reasoning plays an important role in developing and using an ontology. In this layer, we proposed a reasoning mechanism based on SWRL rules to infer instantaneous services and suggest personalized recommendations to each asthma patient. These rules are described in general terms. The combination of ontologies with predefined rules of asthma triggers allows the system to evaluate the risk associated with environmental and food hazards. These rules are specialized for each type of asthma patient according to their medical profile and the expert physician. The semantic formalization of entities associated with the use of a rule-based inference engine allows designing an intelligent environment to improve self-management and to promote real-time protection services against asthma exacerbations. Risk management is the practice of using three major interrelated processes: (1) identifying subjects (patient and allergen); (2) identifying condition indicators (medical profile, location and time) and (3) identifying event patterns, either in the nutritional parameters of the patient or in the surrounding environment. We wrote SWRL rules from valid relationships between asthma concepts to detect allergens and estimate food sensitivities. In total, 46 rules were developed to calculate the allergy risk levels to each

patient, 36 rules to detect the total environmental triggers (Table 2), and 10 rules to avoid common food allergens and prevent a potentially life-threatening reaction (Table 3).

**Table 2.** Environmental rules.

| Rules | Description |
|---|---|
| Rule 1 | To set a maximum level for carbon monoxide (CO) |
| Rule 2 | To set a maximum level for formaldehyde (HCHO) |
| Rule 3 | To set a maximum level for volatile organic compounds (TVOC) |
| Rule 4 | To set a maximum level for carbon dioxide ($CO_2$) |
| Rule 5 | To set a maximum level for particulate matter $PM_{10}$ |
| Rule 6 | To set a maximum level for particulate matter $PM_{2.5}$ |
| Rule 7 | To set a maximum level for ozone ($O_3$) |
| Rule 8 | To set a maximum level for bacteria |
| Rule 9 | To set a maximum level for nitrogen dioxide ($NO_2$) |
| Rule 10 | To set a maximum level for sulfur dioxide ($SO_2$) |
| Rule 11 | To set a maximum level for hydrogen sulfide ($H_2S$) |
| Rule 12 | To set a maximum level for nitric oxide (NO) |
| Rule 13 | To set a maximum level for nitrogen oxides (NOx) |
| Rule 14 | To set a maximum level for total reduced sulphur (TRS) |
| Rule 15 | To set a maximum level for cat dander |
| Rule 16 | To set a maximum level for dog dander |
| Rule 17 | To set a maximum level for horse dander |
| Rule 18 | To set a maximum level for D. farinae |
| Rule 19 | To set a maximum level for D. pteronisius |
| Rule 20 | To set a maximum level for feathers |
| Rule 21 | To set a maximum level for indoor dust |
| Rule 22 | To set a maximum level for grasses |
| Rule 23 | To set a maximum level for ragweed |
| Rule 24 | To set a maximum level for weeds |
| Rule 25 | To set a maximum level for phleola |
| Rule 26 | To set a maximum level for perennial ryegrass |
| Rule 27 | To set a maximum level for tree |
| Rule 28 | To set a maximum level for birch |
| Rule 29 | To set a maximum level for maple |
| Rule 30 | To set a maximum level for oak |
| Rule 31 | To set a maximum level for elm |
| Rule 32 | To set a maximum and minimum level for temperature |
| Rule 33 | To set a maximum and minimum level for pressure |
| Rule 34 | To set a maximum level for windchill |
| Rule 35 | To set a maximum and minimum level for humidity |
| Rule 36 | To set a maximum level for precipitation |

**Table 3.** Food allergy rules.

| Rules | Description |
|---|---|
| Rule 1 | It can be used to rule out an egg allergy |
| Rule 2 | It can be used to rule out a nut allergy |
| Rule 3 | It can be used to rule out a Cladosporium allergy |
| Rule 4 | It can be used to rule out a Hormodendrum allergy |
| Rule 5 | It can be used to rule out a chlado hormod allergy |
| Rule 6 | It can be used to rule out an Alternaria allergy |
| Rule 7 | It can be used to rule out a mixed flour allergy |
| Rule 8 | It can be used to rule out an Aspergillus fum allergy |
| Rule 9 | It can be used to rule out a penicillium allergy |
| Rule 10 | It can be used to rule out a peanut allergy |

An example of risk recognition reasoning based on an SWRL rule is presented in the next flowchart (Figure 4). This example illustrates a further explanation of risk management processes that are based on the subject, condition and event. In this example, we consider an asthma patient from a certain profile is walking outdoors. According to the proposed

method, the patient and environment are the subjects to be tracked, the medical profile and location are classified as condition indicators, while the PM$_{2.5}$ level is a potential risk event. Then, the risk situation is recognized; in this case, the rule refers that there is a serious risk where the level of PM$_{2.5}$ is too high.

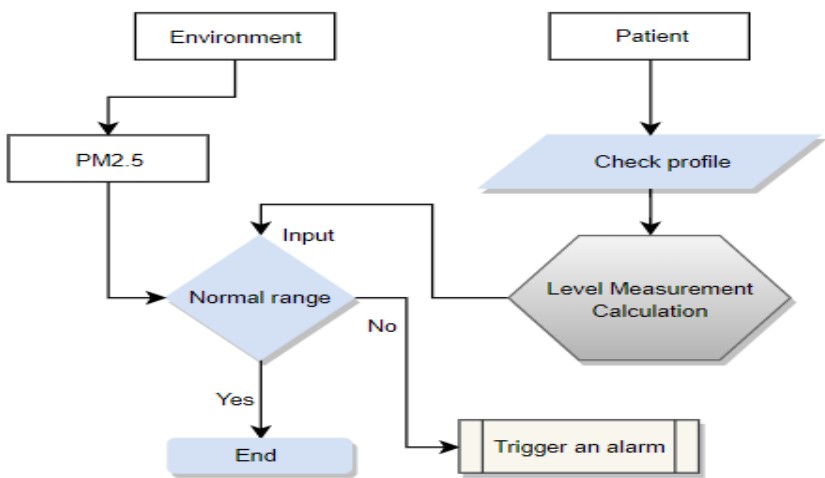

**Figure 4.** An example of risk recognition reasoning based on rules.

This process can be expressed as SWRL rules as follows:

*Patient(?P)ˆEnvironment(?Outdoor)ˆLocatedAt(?P,?Outdoor)ˆAllergen(PM$_{2.5}$)ˆhasCurrent Value(?PM$_{2.5}$,?CVˆ*

*hasMinNormalrange(?CV,?Min)ˆhasMaxNormalRange(CV,Max)ˆ swrlb:*

*greaterThanOrEqual(?CV,Min)ˆswrlb:lessThanOrEqual(?CV,Max)->has_alarm_level(?P, No_Risk)*

In conclusion, the use of SWRL rules in combination with ontology instances has been applied in this work to provide personalized care services to asthma patients. Based on the indoor air quality index, the patient could ask for the suggestions of all the IAQI levels (Table 2). This query could show an aspect of the system in monitoring the environmental conditions. On the other hand, the rules of weather were set to trigger alarms related to temperature, humidity, atmospheric pressure, wind and precipitation. Rules 32 to 36 in Table 2 show the purpose of each rule. The rules for food allergies in the Table 3 alert the patient based on a prior examination about their body's reactions to egg, nuts, Cladosporium, Hormodendrum, chlado hormod, Alternaria, mixed flour, Aspergillus fum, penicillium, and peanuts.

### 3.4. The Application Layer

This patient interface is a personalized platform that helps asthma patients track their symptoms and their daily activities. The asthma navigator helps the patient adhere to treatment plans, prevent exacerbations and avoid allergic reactions.

The risk estimation interface provides more details about the estimated risk of asthma in terms of the total exposure to each factor such as temperature, PCM2.5, dust, CO, etc.

For the food questionnaire interface, the patient is required to answer a set of different questions related to food allergies.

On the other hand, the physician interface provides allergen charts to help healthcare professionals identify patients' conditions with better symptom control and decrease unexpected medical visits.

These interfaces use the different parts given in Section 4, such as the ontology and the rules for asthma. They are used to help patients and to estimate risk factors.

## 4. Implementation

### 4.1. Design of Ontology Model

To implement this asthma monitoring system, we use several tools and different types of methods. The proposed ontology is built using the open-source ontology editor Protégé (Figure 5). The ontology was formalized in OWL semantics and abstract syntax, which is based on a particular version of description logic (DL). We add the class disjoint and characteristics of objects into this ontology.

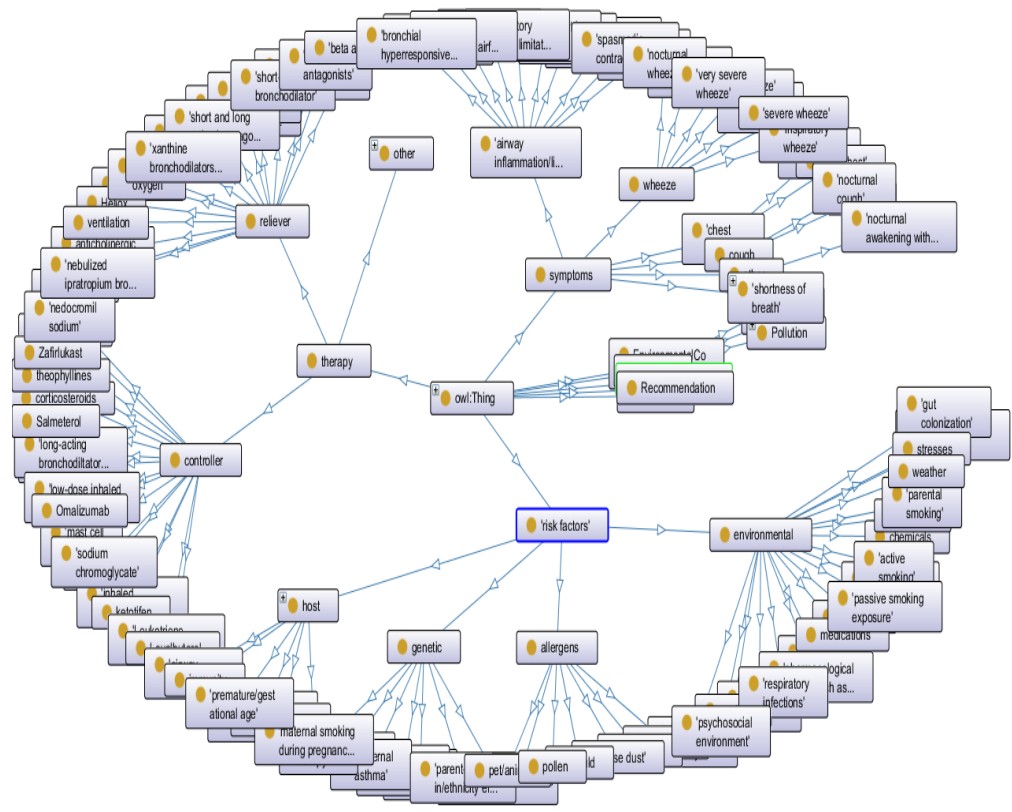

**Figure 5.** Part of proposed ontology.

In contrast, Pellet reasoner was chosen to meet the system requirements. The protection rules were written in SWRL. The SWRLTab plugin in the Protégé 5.5 Editor provides a set of standalone graphical interfaces for managing SWRL rules (Figure 6). This asthma ontology is given in the open access link [41].

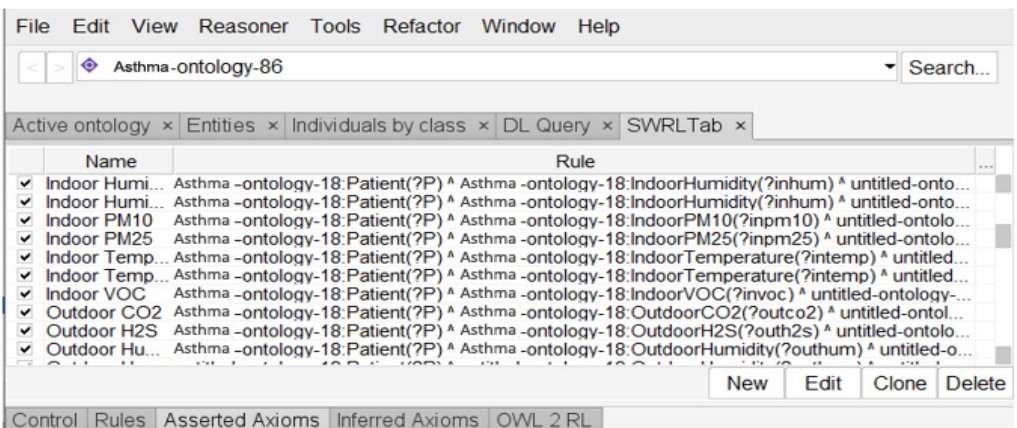

**Figure 6.** SWRL rules in SWRLTab.

To validate this work, in the second part of the project we will create an exposure estimation framework, in which a group of asthma patients will be recruited to accurately test the exposure rules either at home or during travel between locations. In this experimentation, the individual with asthma should agree to keep the environment quality unit for monitoring and storing indoor and outdoor risk factors such as temperature, pressure, humidity and air quality as well as answering questions about their daily life activities on the personalized application interface.

*4.2. Applicability to Other Domains*

To adapt our solution to the monitoring of other chronic diseases, the important steps below must be followed:

- collecting data from asthmatic individual context parameters;
- collecting patient medical profile information;
- identifying and validate medical rules for risk factor control;
- identifying the risk factors of the disease;
- identifying the different recommendations;
- providing disease surveillance services.

The rules will be injected into the reasoning engine of the ontology. The context and patient profile data will be used to instantiate the ontology. The reasoning engine of our solution will then be able to propose services or recommendations according to the patient's context and the rules.

## 5. Conclusions and Future Work

The main contribution of our work is a medical alert system. This system is based on a combination of ontology and SWRL rules to provide various types of protection for asthma patients. The integration of ontologies and medical inference rules is an interesting approach that takes advantage of knowledge representation and reasoning mechanisms. The separated structure of our ontology facilitates the development, maintenance, and validation, as well as the reuse, modification and extension of further domain-specific ontologies. On the other hand, at the top of the representation model, rules-based reasoning is applied to protect patients against risk factors. This approach provides a flexible solution for automating custom monitoring services. This work opens the way to new perspectives that we consider interesting. Returning to the introduction and the proposed goals, it seems clear that there are still many research steps to perform before telemedicine systems are implemented. As a part of these efforts, this research is considered one of the systematic attempts to create an integrated solution for contextual remote medical services based on the context, ontology and rules. This approach needs to be improved by considering the profile of patients and relevant parameters to efficiently execute the needed rules. As future work, we intend to complete the ontology, the rules, then test the proposed rules in a real environment, generalize our ontology to other diseases, create a system compatible with the international standard HL7 and develop a security protocol. More research works will be conducted to accurately test the exposure rules in indoor and outdoor locations. In addition, FAIR principals [42] will be considered to ensure the interoperability and reusability of the different parts of this ontology.

**Author Contributions:** C.L. designed of the asthma cohort and acquisition of data. H.A. and H.M. designed the presented idea. H.A. and H.M. designed the model and analyzed the computational framework of the ontology. H.A. wrote the manuscript in consultation with H.M. and C.L., H.M. and C.L. reviewed and improved the paper. All authors have read and agreed to the published version of the manuscript.

**Funding:** This work was sponsored and funded by the Natural Sciences and Engineering Research Council of Canada (NSERC: RGPIN-2017-05521), and the computer science and mathematics and fundamental sciences departments, and funds FUQAC (2021–2022) of the Université du Québec à Chicoutimi (Quebec), Saguenay, Quebec, Canada. For the asthmatic individual data, the work was

supported by operating grants from the Canadian Institutes of Health Research (CIHR#133605 & #141900; Laprise).

**Data Availability Statement:** Not applicable, the study does not report any data.

**Acknowledgments:** The authors thank the participants for their valuable participation in this study. Catherine Laprise is part of the Quebec Respiratory Health Network (RHN; https://rsr-qc.ca/en/, accessed on 8 June 2022), the Director of the *Centre Intersectoriel en Santé Surable de l'UQAC* (CISD) as for the Canada Research Chair on Genomics of asthma and allergic diseases (http://www.chairs.gc.ca, accessed on 8 June 2022).

**Conflicts of Interest:** The authors declare no conflict of interest associated with this publication.

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
