# Peer review of "First Steps of Asthma Management with a Personalized Ontology Model"

_futureinternet, doi:10.3390/fi14070190_

Round 1

Reviewer 1 Report

The paper presents an ontology-driven framework supported by Semantic Web Rule Language (SWRL) medical rules. According to the Authors, this ontology-based approach is aimed to provide a personalized care for an asthma patient by identifying the risk factors and the development of possible exacerbations. As the Authors claimed, the originality of the proposed approach resides in the dynamic control of thresholds that govern the triggers of asthma. The topic is interesting and the paper is well corresponding to the journal aim and scope.

This paper is resubmitted.

The Authors finally decided to add the missing ontology. The SWRL rules are implemented. The ontology (OWL file) works, but there are shortcomings, for example, the Authors did not make all primitive siblings disjoint. Next, object properties are not complete (the characteristics are missing).

Overall the idea for the article is good, but the presentation of the research should be improved.

Minor typos:

Figure 5 is illegible

Author Response

Thank you for your comments. We update now the ontology to include class disjoint and characteristics of objects. This ontology-driven framework based on the medical rules is published online for open access and it could help the community of researchers/practitioners. Thank you for your collaboration and your effort to evaluate our work.

Reviewer 2 Report

The paper proposes a solution that is technically sound and sufficiently original. The paper is adequately well structured and organized.

I already reviewed the previous version of the paper by stating that there was the important lack of a reference to the proposed ontology, so that the community of researchers/practitioners in the field could benefit from its open availability in their research work. Now, this lack has been solved.

Therefore, at this stage, I recommend accepting the paper.

Only a few writing inaccuracies and typos have to be fixed when preparing the camera-ready version of the manuscript. For example:

- "state of the arts" -> "state of the art"

- "future works" -> "future work"

Author Response

Thank you very much for your comments. We take care of the inaccuracies and typos and improve our camera-ready version.